# Seroprevalence and risk factors associated with bovine tuberculosis in cattle in Eastern Bhutan

**Karma Wangmo**[1]*, **Ratna B Gurung**[2], **Tshering Choden**[3], **Sangay Letho**[1], **Narayan Pokhrel**[4], **Lungten Lungten**[1], **Tashi Zangmo**[1], **Sonam Peldon**[2], **Kinzang Chedup**[5], **Sylvia Jaya Kumar**[6], **Thinley Dorji**[7], **Sangay Tshering**[8], **Kinzang Dorji**[9], **Tenzin Tenzin**[2]

**1** Regional Livestock Development Centre, Department of Livestock, Khangma, Trashigang, Bhutan, **2** National Centre for Animal Health, Department of Livestock, Thimphu, Bhutan, **3** District Veterinary Hospital, Department of Livestock, Lhuentse, Bhutan, **4** District Veterinary Hospital, Department of Livestock, Mongar, Bhutan, **5** District Veterinary Hospital, Department of Livestock, Samdrup Jongkhar, Bhutan, **6** Loyola College, University of Madras, Chennai, India, **7** Kanglung Hospital, Ministry of Health, Trashigang, Bhutan, **8** Trashiyangtse hospital, Ministry of Health, Trashiyangtse, Bhutan, **9** Eastern Regional Referral Hospital, Ministry of Health, Mongar, Bhutan

* karmavet2011@gmail.com

**Data Availability Statement:** The authors confirm that all data underlying the findings are fully available without restriction. All relevant data are within the paper.

## Abstract

Bovine tuberculosis (bTB) is a chronic zoonotic disease affecting cattle of all age groups including wild animals. It poses a significant threat to public health and high economic losses to dairy farmers. While the disease has been eradicated from most of the developed countries through extensive surveillance, testing and culling strategy, it is endemic in Africa, Asia, and the Middle East countries. Currently, there is limited research regarding the prevalence of bTB in cattle in Bhutan. This study aimed to determine the seroprevalence of bTB in cattle in six districts of eastern Bhutan. A two-stage probability proportional to size (PPS) sampling strategy was used to determine the number of animals from which serum samples needed to be collected in each district and sub-district. All farms and cattle for sampling were randomly selected from the data in the annual livestock census of 2020. The samples were tested using bTB ELISA test kit. The seroprevalence and their 95% confidence intervals were calculated. Logistic regression models were constructed to assess the influence of various individual animal and environmental risk factors (breed, age, sex, source of animal, body condition scores of animals, respiratory system status) associated with sero-positivity in animals.

The study revealed an apparent seroprevalence of 2.57% (25/971 cattle; 95% CI:1.58–3.57), with an estimated true seroprevalence of 0.91% (95% CI: 0.0–2.81). However, none of the variables were found to be significantly associated with bTB seroprevalence in cattle.

We recommend, further sampling and employment of confirmatory testing to fully ascertain the extent of bTB in the cattle herds in eastern Bhutan for prevention and control.

**Funding:** The author(s) received no specific funding for this work.

**Competing interests:** 'The authors have declared that no competing interests exist.

## Author summary

Bovine tuberculosis (bTB) is a bacterial zoonotic disease caused by *Mycobacterium bovis*, affecting cattle of all ages. *Mycobacterium bovis* imposes significant economic burdens on livestock farmers, primarily through reduced production, trade restrictions for live animals and animal products, and the costs of control measures. The disease can be transmitted to humans via consumption of raw milk or through inhalation of infected aerosols. Worldwide, the bovine tuberculosis accounts for 10% of the human tuberculosis cases. In Bhutan, the prevalence and incidence of bTB in both cattle and humans are unknown. In this study, we estimated the seroprevalence of bTB among cattle in six eastern districts of Bhutan. We collected serum samples from 971 cattle and tested them for bovine tuberculosis using ELISA. Our study demonstrated an apparent prevalence of 2.57% (95% CI:1.58–3.57) and a true prevalence of 0.91% (95% CI: 0.0–2.81). Additionally, we assessed factors such as age, breed and farm location that could affect the prevalence of bTB, but none of these factors were significantly associated with the prevalence of bTB. Our study revealed a very low seroprevalence of bTB, among cattle in eastern Bhutan. The findings from this study will contribute to design prevention and control measures for bTB in cattle in Bhutan.

## Introduction

Bovine tuberculosis (bTB) is a chronic zoonotic infection caused by Mycobacterium tuberculosis complex, primarily by *Mycobacterium bovis (M.bovis)*. While the disease predominantly affect cattle, other domestic animals, some species of wild animals, and even humans are susceptible [1]. It is a progressive disease with significant socio-economic and public health impacts. The disease mainly infects lungs and other organs in later stages, with progression occurring over a period ranging from months to years in some cases [1]. In addition, latent infections and anergic reactors pose challenges in detecting and preventing bTB [2].

Economic losses due to bTB infections are primarily related to reduced production, trade restrictions for live animals and animal products, and the costs of control measures including testing and culling of animals [3]. *Mycobacterium bovis* imposes significant economic burdens on livestock farmers, with an estimated over 50 million cattle being infected worldwide, resulting in annual costs exceeding USD 3 billion [4,5]. Besides, the disease also accounted for 10% of human TB cases with 147,000 new cases of zoonotic TB in humans and 12,500 deaths in 2016 alone [6]. Humans may contract bTB by consuming raw milk or by inhaling contaminated aerosols when in close contact with infected animals [7,8]. Before the implementation of mandatory pasteurization in many countries, M. bovis was responsible for approximately 25% of tuberculosis (TB) cases in children [4,5].

Bovine tuberculosis is found globally, with the highest occurrence observed in countries across Africa and Asia [9]. In 2018, 94 out of 182 World Organisation for Animal Health (WOAH) listed countries reported the presence of bTB in domestic animals and in wildlife [3]. The prevalence of bTB in cattle has been reduced to a negligible level in most developed countries through extensive test and slaughter methods [10,11]. However, the disease remains endemic and uncontrolled in Africa, Asia, Latin America and most countries in the Middle East [12–15] posing great economic and public health threat. High prevalence of bovine tuberculosis (bTB) were reported in several countries including 19% in Ghana (2011–2012) [16], 28% in South Africa (2016–2017) [12] 22% in Ethiopia (2016–2017) [13], 45.6% in Bangladesh (2019) [17] and 7.3% in India (2018) [18]. The prevalence of bTB is influenced by risk factors

such as individual animal level, herd level, and wildlife reservoirs [19]. Individual risk factors include age, breed, sex, body condition, immune status, and genetic resistance of the animal [20,21]. The herd level risk factors include history of bTB outbreak in the herd or in the household, herd size, farm-biosecurity and type of cattle management [22]. Older cattle were found to be at higher risk of contracting bTB compared to yearlings and calves [17,23]. Breed variation has been observed in susceptibility to bTB with some studies indicating exotic breeds being more susceptible to bTB compared to indigenous breeds. With respect to sex, some studies found female cattle to be more susceptible whereas some findings suggest it otherwise [14,23,24]. The prevalence of bTB is also dependent on farm density in an area, distance between two farms and also type of farming—stall feeding or pasture grazing [24]. Moreover, the risk of transmission also increases with the increase in herd size [25]. Additionally, wildlife serves as a great source of bTB in many parts of the world hindering bTB control measures [15]. This is more so in the shared pastures where cattle from different farms and wildlife can access [19].

Single intradermal tuberculin test (SIT) with a sensitivity of 92% and specificity of 89% [26] is commonly used test to identify positive reactors. Other tests such as comparative intradermal test (CIT), interferon gamma (IFN- γ) assay, rapid antigen and enzyme linked immunosorbent assay test (ELISA) are also used to detect bTB [14,16]. SIT, CIT and IFN- γ tests cell mediated immune response. SIT and CIT requires two field visits, restrain of animal and injection of bTB antigen. IFN- γ requires time limited cell sensitisation, incubation period and blood sample collection. In contrast, the ELISA is logistically easier to perform and provides standardised interpretation and exhibits higher specificity (96%) compared to the SIT [14].

Bovine tuberculosis has not been detected in Bhutan [27]. However, bTB is highly prevalent in neighbouring countries of India, Bangladesh, and Pakistan. In India, the bTB prevalence varied from 40% in Assam in 2014 [28] to over 25% in West Bengal in 2017 [29], which shares a border with Bhutan. A recent study in Bangladesh in 2019 revealed an alarming overall bTB prevalence in the region, reported at 45.6% [17]. Bhutan has recently imported live cross-bred cattle from India to enrich its genetic diversity, inadvertently raising the risk of introducing infectious diseases such as bTB [27]. Furthermore, there has been a shift in Bhutan's cattle rearing practices from extensive free grazing to more intensive indoor stall-feeding systems [30], which may contribute to increased transmission of bTB. Therefore, it is imperative to assess the disease status among cattle in Bhutan to institute appropriate control and preventive measures. Accordingly, this study aimed to determine the seroprevalence of bovine tuberculosis in cattle in eastern Bhutan and identify the associated risk factors for seropositivity.

## Methodology

### Ethics statement

The study was approved by the Research and Extension Division of the Department of Livestock, Ministry of Agriculture and Forest, Bhutan on 16 February 2021 via letter number DoL/RED-142/2020-21/1774.

### Study area

Bhutan is administratively subdivided into 20 districts. Its veterinary and livestock services are organized into four regions, each overseen by a Regional Livestock Development Centre (RLDC) located in the East, West, East Central, and West Central at the time of conducting this study in 2021. This study was conducted within the eastern RLDC region, which provides services to six districts: Trashigang, Trashiyangtse, Lhuentse, Pema Gatshel, Mongar, and Samdrup Jongkhar (Fig 1). Notably, four of these districts share borders with India—

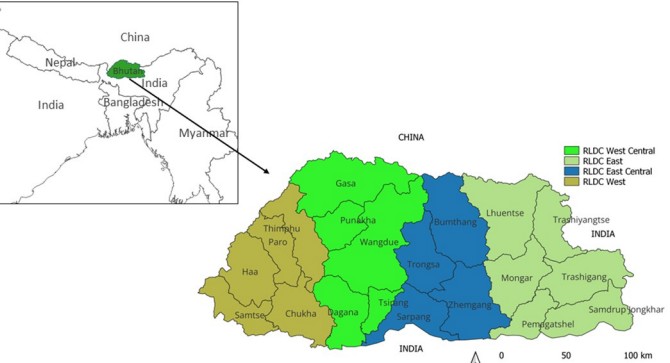

**Fig 1. A Bhutan map showing four regions for veterinary and livestock service delivery purposes.** The bTB study was conducted in the Regional Livestock Development Centre (RLDC East region) area that comprises of six districts. Bhutan borders with China in the north and India in the south, southeast, and south-west. The basemap shapefiles for Bhutan and the world were downloaded from an open-source, the DIVA-GIS website (https://www.diva-gis.org/). The map was created in a free and open-source software program QGIS (https://www.qgis.org/en/site/).

Trashigang and Trashiyangtse with Arunachal Pradesh State, and Pema Gatshel and Samdrup Jongkhar with Assam State. Due to financial and logistical constraints, the study's scope was limited to this specific region.

Different cattle rearing systems are practiced based on the area's topography. In the higher altitude regions such as Trashigang, Trashiyangtse, and Lhuentse, a transhumant system is practiced, wherein cattle migrate seasonally. In the lower altitude areas like Mongar, Pemagatshel, and Samdrup Jongkhar, a sedentary system is practiced [31]. In the sedentary system, indigenous cattle are allowed to freely graze on shared pastures and watering points, while high yielding crossbred dairy cattle are stall-fed [30].

## Study design

This was a cross-sectional study conducted to determine the seroprevalence of bTB, from March 2021 to November 2021. Data on animal age, sex, breed, and source of cattle (imported or farm born) were collected using a semi-structured questionnaire. Additionally, basic inquiries regarding the respiratory system, such as nasal discharge, coughing status, and unusual sounds observed, were also included to correlate apparent clinical signs of bTB with each animal.

## Sample size and sampling strategy

During the study period, six eastern districts have a cattle population of 96,581 (Mongar district: 26,977; Trashigang: 25,532; Lhuentse: 13,933; Samdrup Jongkhar: 12,191; Tashiyangtse: 10,150; Pemagatshel: 7,79) [32]. The sample size was calculated using the EPITOOLS [33,34] with a 95% confidence interval and a precision of 0.05. The bTB testing was performed using WOAH approved ELISA kit (Antibody Test Kit from IDEXX [35] with a diagnostic sensitivity of 65% and specificity of 98% [35]. Due to the absence of bTB prevalence data in Bhutan or in eastern Bhutan, a prevalence of 50% was assumed to maximize the sample size, requiring 863 cattle to be sampled from a total of 96,581 for this study. Ultimately, 971 samples were collected to increase the power of the study and to account for potential damage or loss of samples.

A two-stage probability proportional to size (PPS) sampling strategy was used to determine the number of samples to be collected from each district and sub-district. The calculation was preformed using Microsoft Excel (Microsoft Excel, Redmond USA) with a district cattle data used for selection [32]. The number of samples to be collected from each district was determined based on the cattle population of that district. Similarly, the cattle population of each sub district and the total sample of that district was used to determine the total samples to be collected from each sub district. All household owning cattle in each sub-district were assigned a random number to select the households. In cases where households had more than one cattle, the animal for sampling was chosen using simple random sampling to prevent any potential selection bias in the study.

## Sample collection

Sample collection was carried out from March 2021 to November 2021. A verbal informed consent was obtained from the cattle owners selected for the study, and all have consented to sample collection and interview. Approximately 6–7 mL of blood via jugular venepuncture into plain vacutainers (BD, India) was collected. After collection, blood samples were allowed to settle for 10 to 15 minutes to facilitate clotting and serum separation before moving on to the next farm. Each sample from a cow was assigned a sample identification number based on the order of sampling. Blood samples were collected in the morning and evening; either before the cattle were released for grazing or upon their return to the shed in the evening. Serum samples were processed within 4–5 hours of sample collection on all occasion. Blood samples were centrifuged at 1000 x g for 15 to 20 minutes, and the serum was transferred to well- labelled screw-capped cryovials for storage at -20˚C [36]. The frozen serum samples were then shipped to Regional Livestock Development Centre's Veterinary Laboratory, Khangma, Trashigang for testing.

## Laboratory analysis

The frozen serum samples were thawed and tested using the *Mycobacterium bovis* Antibody Test Kit bovine (IDEXX, *M bovis*) as per the manufacturer's protocol [35].

## Data analysis

The data was analysed using R package version 4.1.2 and Microsoft Excel (Microsoft Excel, Redmond USA). The demographic and clinical data are presented in frequency and proportions. The apparent prevalence (number of animals samples tested seropositive to the test by the total animals or samples tested) and the 95% confidence intervals were calculated [37]. The diagnostic sensitivity (Se) and the specificity (Sp) of bTB testing ELISA kit that was used for this study was 65% and 98%, respectively [35]. Since we have used a test with less than 100% sensitivity and specificity, the estimated prevalence would be invariably biased which makes the estimate of the prevalence incorrect. Using the known test sensitivity and specificity, it can calculate this inherent error of the test, and convert the results of the analysis, the apparent prevalence (AP), to the corrected result, the true prevalence. Using the estimated sensitivity and specificity of the test, we estimated an approximately unbiased true prevalence (TP) of seropositive cases in a population using Rogan-Gladen estimater (1978): *TP = AP + (Sp-1) / Se + Sp-1)*. The variance for the true prevalence was estimated using Rogan-Gladen estimater (1978) [38] and Greiner and Gardner (2000) [37]: *Var (TP) = AP (1-AP) / nJ²*, where *n* is the total sample size of the prevalence study and *J* is the Youden index (J = Se+Sp-1) [39]. Using these results, the simplest, so-called Wald confidence limits for the true prevalence based on normal approximation was calculated using the formula [40] *TP ± Z$_{crit}$ * var (TP)$^{1/2}$*. Here *Z$_{crit}$*

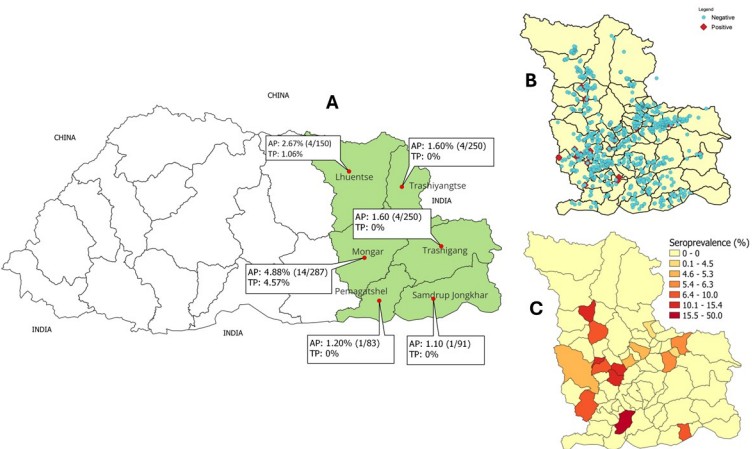

**Fig 2. Apparent prevalence (AP) and True prevalence (TP) of bTB in cattle in six eastern districts are shown on the map (A).** The sampling sites and test results (B) and seroprevalance of bTB in cattle at the sub-district level are also shown on map (C). The basemap shapefiles for Bhutan were downloaded from an open-source, the DIVA-GIS website (https://www.diva-gis.org/). The map was created in a free and open-source software program QGIS (https://www.qgis.org/en/site/).

is the critical value of the standard normal distribution belonging to the prescribed confidence level which is 1.96. However, some authors recommend Blaker's interval for general use and thus Blaker's 95% confidence interval for the TP was calculated and derived using the EPI-TOOLS [41].

The cattle demography, source of the cattle, and some of the pertinent clinical signs of bTB (cough, nasal discharge, respiratory distress) were included as explanatory variables against the binary outcome variable of a sample being seropositive or negative for the test. Fisher's exact test and Chi square test were performed to identify the variables that were significantly associated with bTB seropositivity. Largely there were two types of breeds in this study: cross-bred and indigenous. Age was categorized as young (<4 years), adult (4–8 years), and old (>8 years). Cattle sources classified into four categories: imported from other country (India), purchased from other districts, purchased within district, and bred within the farm. A five-point scale was used to estimate the body condition score (BCS) of the cattle [42], with categories of 1–2 for thin, 3–4 for average, and >4 for fat cattle. The clinical signs of cattle such as cough, nasal discharge and respiratory distress observed by the owners were categorised based on frequency of occurrence: "No" for no occurrence, "Occasionally" for occasional occurrence and "Frequently" for frequent occurrence.

## Results

A total of 971 serum samples were collected from randomly selected cattle across six eastern districts of Bhutan (Fig 2). The maximum number of samples were collected from Mongar district considering its large cattle population. Details of the number of samples collected and tested from each district is provided in in Tables 1 and 2.

More than 90% (n = 899) of the study animals were female, with 71% (n = 697) of them being crossbred cattle (Table 1). The median age of animal was 5 years (IQR 3 years), with significant differences observed between male and female cow (Wilcoxon test p-value <0.001). There were no significant differences in bTB cases by age groups (p-value 0.283). Of the total, 52.3% (n = 513) were farm born, while 2.8% of them were imported from India. Most of the

**Table 1. Risk factors associated with the sero-positivity of bTB in cattle in six- districts of eastern Bhutan.**

| | Negative (N = 946) | Positive (N = 25) | P-value |
|---|---|---|---|
| **Districts** | | | |
| Lhuentse | 146 (15.4%) | 4 (16.0%) | 0.153 |
| Mongar | 273 (28.9%) | 14 (56.0%) | |
| Pemagatshel | 82 (8.7%) | 1 (4.0%) | |
| Samdrup Jongkhar | 90 (9.5%) | 1 (4.0%) | |
| Trashigang | 246 (26.0%) | 4 (16.0%) | |
| Trashiyangtse | 109 (11.5%) | 1 (4.0%) | |
| **Breed** | | | |
| Crossbred | 679 (71.8%) | 18 (72.0%) | 1 |
| Indigenous | 267 (28.2%) | 7 (28.0%) | |
| **Age (years)** | | | |
| <4 | 351 (37.1%) | 6 (24.0%) | 0.283 |
| 4–8 | 473 (50.0%) | 14 (56.0%) | |
| >8 | 122 (12.9%) | 5 (20.0%) | |
| **Sex** | | | |
| Female | 875 (92.5%) | 24 (96.0%) | 1 |
| Male | 71 (7.5%) | 1 (4.0%) | |
| **Source** | | | |
| Imported | 26 (2.7%) | 2 (8.0%) | 0.0992 |
| Other districts | 109 (11.5%) | 0 (0%) | |
| Farm born | 498 (52.6%) | 15 (60.0%) | |
| Within district | 313 (33.1%) | 8 (32.0%) | |
| **BCS** | | | |
| 1–2 (thin) | 279 (29.5%) | 5 (20.0%) | |
| 3–4 (average) | 645 (68.2%) | 20 (80.0%) | |
| >4 (fat) | 22 (2.3%) | 0 (0%) | 0.592 |
| **Cough** | | | |
| Frequently | 52 (5.5%) | 2 (8.0%) | 0.239 |
| Occasionally | 142 (15.0%) | 6 (24.0%) | |
| No | 752 (79.5%) | 17 (68.0%) | |
| **Respiratory distress** | | | |
| Frequently | 21 (2.2%) | 0 (0%) | 0.692 |
| Occasionally | 78 (8.2%) | 3 (12.0%) | |
| No | 847 (89.5%) | 22 (88.0%) | |
| **Nasal Discharge** | | | |
| Frequently | 3 (0.3%) | 0 (0%) | 0.736 |
| Occasionally | 74 (7.8%) | 1 (4.0%) | |
| No | 869 (91.9%) | 24 (96.0%) | |

cattle (68.5%, n = 665) had an average body condition score of 3–4, with only 2.2% (n = 22) categorised as fat (BCS >4).

Of the 971 cattle, cattle owners reported that 5.5% (n = 54) of the cattle coughed frequently while 2.2% (n = 21) of the cattle had frequent respiratory distress, and 0.3% (n = 3) of the cattle had frequent nasal discharge.

The apparent seroprevalence of bTB was 2.6% (n = 25) (95% CI:1.6–3.6). Based on this, the true seroprevalence was calculated to be 0.91% (95% CI: 0.0–2.8). Highest true prevalence was

**Table 2. Sero-prevalence of bTB in cattle in six-districts of eastern Bhutan.**

| Districts | Apparent prevalence (%(n)) | 95% CI | True prevalence (%) | 95%CI | Sample collected/Cattle population |
|---|---|---|---|---|---|
| Lhuentse | 2.67(4/150) | 0.04–5.24 | 1.06 | 0.0–7.39 | 146/13943 |
| Mongar | 4.88 (14/287) | 2.39–7.37 | 4.57 | 1.47–9.56 | 287/26977 |
| Pemagatshel | 1.20 (1/83) | 0.0–3.55 | 0.0 | 0.0–7.16 | 83/7798 |
| Samdrup Jongkhar | 1.10 (1/91) | 0.0.-3.55 | 0.0 | 0.0–6.29 | 88/12187 |
| Trashigang | 1.60 (4/250) | 0.42–3.16 | 0.0 | 0.0–3.24 | 250/25532 |
| Trashiyangtse | 0.91 (1/110) | 0.0–2.68 | 0.0 | 0.0–4.72 | 110/10150 |

recorded in Mongar district at 4.57% (Fig 2), followed by Lhuentse at 1.06% (Table 2). The true prevalence was zero in the remaining four districts (Table 2). We also fitted the demographic and clinical data to multiple logistic regression and analysed for confounding factors using backward elimination (S1 Table).

## Discussion

Our study aimed to determine the seroprevalence of bovine tuberculosis in six dairy hub districts of Eastern Bhutan. The overall apparent seroprevalence of bTB was 2.6%, with a true seroprevalence of 0.91%. The low seroprevalence may be attributed to limited exposure of the cattle population to the bTB-causing pathogen, *Mycobacterium bovis*. Factors such as low cattle population density, extensive free grazing as part of the cattle management system, scattered locations of villages and farms in the study area restricting cattle mixing, and the geographical terrain of the country (high mountains) may have contributed to reduced exposure to the pathogen. There is limited mixing of cattle from this study area with neighbouring countries such as India, where bTB prevalence in cattle is higher [28, 29]. Although live cattle were imported from India to enrich genetic diversity and enhance milk production in Bhutan, only healthy cattle from government-recognized farms were purchased, thereby minimizing the risk of introducing *Mycobacterium bovis* [43]. Only two cattle who tested seropositive to bTB was imported from India. Furthermore, we have not detected any clinical cases or pathological lesions consistent with bTB in cattle or other animals in Bhutan [27]. However, it is possible that sporadic cases of bTB may be present in the cattle population, resulting in low overall prevalence. Without active surveillance and diagnostic testing, these cases may go undetected.

It is also important to acknowledge that the lower seroprevalence detected in this study may be influenced by the choice of test method employed. *Mycobacterium bovis* antibody ELISA test kit used in this study has a sensitivity of 65% (59.7%–69.5%), notably lower than the intradermal tuberculin test commonly used for bTB detection [44]. Furthermore, the sensitivity of ELISA test kit has been reported to be considerably lower than standard intradermal skin test [2] with field-level sensitivity ranging from 4–9% compared to the intradermal test and 9–14% compared to post-mortem techniques [2]. However, the sensitivity of ELISA kit increases to 66–85% when tested on samples from post-skin test in the same cohort [45]. Use of only one test with lower sensitivity in a tuberculin unsensitised population may result in missed bTB positive cases detection. Additionally, antibody tests like ELISA are ineffective in detecting sub-clinical infections, as their sensitivity increases only when infections progress and higher antibody levels are produced [46]. However, the ELISA test specificity of 98% that we have used is comparable to the gold standard culture test (99.1%), minimizing false positives [47]. Although the ELISA test is recommended for use alongside intradermal tuberculin tests for screening purposes, the latter was not utilized in this study to avoid introducing TB

antigen into naive cattle herds [44], and due to logistical constraints. Therefore, screening animals using antibody-detecting test kits was deemed the most feasible approach for determining baseline seroprevalence in this study.

Moreover, use of the ELISA test for bovine tuberculosis (bTB) that have a sensitivity of 65%, indicates that approximately 35% of true seropositive cases may be missed. This presents a risk of false negatives, where infected cattle may go undetected due to factors such as infection stage and antibody level fluctuations. As a results, there is a potential for underestimating bTB seroprevalence when mainly relying on ELISA testing. Additionally, the test's specificity of 98% implies a 2% risk of false positives, attributing negative cases as positive. False positives may arise from cross-reactivity or testing errors. In summary, while ELISA testing provides valuable insights, confirming seropositive results with complementary methods, such as intradermal tuberculin tests is necessary for ensuring accurate diagnosis and surveillance efforts. This approach helps to minimize the risk of misinterpretation of results and enhances overall detection accuracy. This is the main limitation of the ELISA test and complement it with other diagnostic methods to improve the accuracy of bTB detection and surveillance efforts. Therefore, it is important to acknowledge this limitation of the ELISA test that we have used in this study.

The low seroprevalence of bovine tuberculosis (bTB) observed in our study should not be directly compared with the high detection rates reported in other studies. For example, in Assam State, India, which shares a border with one of the districts (Samdrup Jongkhar) of eastern Bhutan, a prevalence of 28% was reported, with tuberculin tests conducted on suspected bTB cattle [28]. Similarly, in Nepal, bTB prevalence ranged from 11% [48] using single intradermal tuberculin tests to 45.7% [49] using tuberculosis antibody test kits, with again samples collected from suspected bTB cattle. It is important to note that the tuberculin test measures cell-mediated immune responses, whereas our study utilized an ELISA kit to measure humoral immune responses [50]. In contrast, a large-scale study conducted in Bangladesh, involving 1865 cattle from 79 randomly selected farms, reported a prevalence of 45.6% [17], significantly higher than the prevalence found in our study, suggesting that bTB is highly endemic in that region.

In our study, we have recorded a difference in seropositivity in relation to breed, sex, age, source of animals, body condition scores and owner reported clinical signs shown by the animals (see Table 1). A higher number of seropositive cases were recorded in crossbred cattle (18/25) compared to indigenous breeds, which could be due to the genetic composition of the indigenous breed that may have higher resistance to infection [14]. More seropositive cases (24/25) detected in females which could be due to female cattle being more susceptible to tuberculosis infection or this could be an artifact of sampling bias since more number of samples are being collected from females especially in dairy herds [12]. Similarly, more seropositive cases (14/25) were detected in adult cattle (4–8 years) compared to young which could be attributed to several factors, and adult cattle are more likely to have been exposed to the bTB-causing pathogen, M. bovis, over a longer period compared to younger animals. Additionally, as cattle age, their immune systems may become less effective at controlling infections, potentially leading to higher rates of seropositivity in older animals. Furthermore, the higher number of positive cases (20/25) were detected in cattle with average body condition scoring [3–4] compared to thin [1–2] and fat (>4). Although this finding is statistically not significant in this study, it is consistent with findings from other studies where cattle with good body condition had higher positivity than those cattle that were thin and emaciated [51]. Overall, these findings suggest that factors such as breed, age and body condition may influence susceptibility to bTB infection in cattle. Further research is needed to better understand the underlying mechanisms driving these associations and to develop targeted interventions for bTB control and prevention in cattle populations. We have also detected more positive cases (15/25) in the farm born cattle compared to cattle introduced from other sources. Surprisingly, only two seropositive cases were

detected in cattle that was imported from India, which is in contrast with findings from other studies where introduction of new cattle from other source increased risk of bTB in the farm [52]. However, this finding should be interpreted cautiously since only few numbers of imported animals were sampled and tested in our study because of random sampling. It could also depend on the actual source of cattle, particularly if the imported cattle were originated from a region known to be either free of bTB or endemic to the disease [53]. As observed in other studies, testing only suspect cases or focusing on high-risk groups of cattle, such as imported ones, may yield different results [28, 48, 50, 54] and we recommend for further risk-based study in future in the country. In addition to the demographic factors, few data on clinical signs related to bTB were also collected and analysed. Cough, nasal discharge, and respiratory distress were statistically not associated with seropositivity of bTB or detection of more positive cases. Although chronic cough, nasal discharge and respiratory distress are some of the signs associated with bTB [55], none of these signs were associated for bTB seropositivity in the study. However, we could not follow up with the seropositive of the cases due to logistical challenges, and a further study is necessary to understand the epidemiology of bTB in cattle in the country. We have discussed and recommended the further line of action in the conclusions.

## Conclusion

Based on the findings and conclusions of this study, we recommend the following:

- Implement comprehensive surveillance programs using more sensitive diagnostic tests to accurately determine the prevalence and distribution of bovine tuberculosis (bTB) in Bhutan. This includes regular testing of cattle populations across different regions of the country. A risk-based surveillance strategy focusing on cattle along the international border, imported animals, commercial dairy farms and those showing clinical signs suggestive of bTB may be prioritized to optimize surveillance efforts and manage costs effectively.

- Invest in training programs for veterinary professionals and laboratory personnel to improve their capacity for bTB detection, diagnosis, and control measures. This will ensure timely and effective responses to suspected cases.

- Educate cattle owners, herders, and the general population about the risks associated with bTB transmission, and preventive measures, such as farm biosecurity, regular testing of cattle against bTB, proper hygiene practices, boiling/ pasteurization of milk for human consumption, to minimise the spread of bTB and other zoonotic diseases.

- Strengthen One Health collaboration by facilitating information sharing between animal and human health sectors, particularly regarding human TB and bovine TB cases, to enhance surveillance and response capabilities.

- Support research and development initiatives aimed at understanding the epidemiology and transmission dynamics of bTB in cattle and risk of human infection in Bhutan. Foster collaboration with international organizations and research institutions to share knowledge, resources, and best practices for bTB control and prevention.

- Develop and implement policies and regulations for the prevention, control, and management of bTB in Bhutan based on a clear understanding of the diseases' epidemiology in cattle in the country. This may involve enforcing Livestock Act and Livestock Rules and Regulation of Bhutan, while incorporating updated strategies and guidelines for bTB control based on WOAH standards [56].

## Supporting information

**S1 Table. Factors associated with bovine TB in Eastern Bhutan.**
(DOCX)

**S1 Data. Raw dataset of the study.**
(XLSX)

## Author Contributions

**Conceptualization:** Karma Wangmo, Thinley Dorji.

**Data curation:** Karma Wangmo, Tshering Choden, Sangay Letho, Sylvia Jaya Kumar.

**Formal analysis:** Karma Wangmo, Tshering Choden, Sangay Letho, Sylvia Jaya Kumar, Kinzang Dorji, Tenzin Tenzin.

**Investigation:** Karma Wangmo, Tshering Choden, Lungten Lungten, Kinzang Chedup.

**Methodology:** Karma Wangmo, Ratna B Gurung, Narayan Pokhrel, Tenzin Tenzin.

**Project administration:** Karma Wangmo, Sangay Letho, Tashi Zangmo, Sonam Peldon.

**Resources:** Karma Wangmo, Ratna B Gurung, Tashi Zangmo.

**Software:** Karma Wangmo, Sangay Tshering, Kinzang Dorji.

**Supervision:** Karma Wangmo, Ratna B Gurung, Tenzin Tenzin.

**Validation:** Karma Wangmo, Narayan Pokhrel.

**Visualization:** Sonam Peldon, Thinley Dorji.

**Writing – original draft:** Karma Wangmo.

**Writing – review & editing:** Karma Wangmo, Ratna B Gurung, Tshering Choden, Sangay Letho, Narayan Pokhrel, Lungten Lungten, Tashi Zangmo, Kinzang Chedup, Sylvia Jaya Kumar, Thinley Dorji, Sangay Tshering, Kinzang Dorji, Tenzin Tenzin.

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
