## [Decision Letter · Decision Letter 0]

26 Feb 2024

Dear Dr Wangmo,

Thank you very much for submitting your manuscript "Seroprevalence and risk factors associated with bovine tuberculosis in cattle in Eastern Bhutan" for consideration at PLOS Neglected Tropical Diseases. As with all papers reviewed by the journal, your manuscript was reviewed by members of the editorial board and by several independent reviewers. In light of the reviews (below this email), we would like to invite the resubmission of a significantly-revised version that takes into account the reviewers' comments. 

We cannot make any decision about publication until we have seen the revised manuscript and your response to the reviewers' comments. Your revised manuscript is also likely to be sent to reviewers for further evaluation.

Sincerely,

Husain Poonawala

Academic Editor

Stuart Blacksell

Section Editor

Reviewer's Responses to Questions

**Key Review Criteria Required for Acceptance?**

**Methods**

-Are the objectives of the study clearly articulated with a clear testable hypothesis stated?

-Is the study design appropriate to address the stated objectives?

-Is the population clearly described and appropriate for the hypothesis being tested?

-Is the sample size sufficient to ensure adequate power to address the hypothesis being tested?

-Were correct statistical analysis used to support conclusions?

-Are there concerns about ethical or regulatory requirements being met?

Reviewer #1: Yes the objectives of the study are clear, design is appropriate, sample size is sufficient, statistical analysis were adequate and ethical statement is being presented

Reviewer #2: The Material and Methods section must be re-structured. The population is not clearly described, data on the population is missing, and it is highly relevant to make inferences and to understand how prevalences are estimated.

 It is long and reiterative. Some data may be ommited or moved to Supplementary material (i.e., laboratory analysis).

**Results**

-Does the analysis presented match the analysis plan?

-Are the results clearly and completely presented?

-Are the figures (Tables, Images) of sufficient quality for clarity?

Reviewer #1: Yes the analysis, results and figures are appropriate

Reviewer #2: The results section should also be re-structured. Descriptive data should be included. Results of the logistic regression analysis (univariable and multivariable) are missing. The order of the Tables should be changed.

**Conclusions**

-Are the conclusions supported by the data presented?

-Are the limitations of analysis clearly described?

-Do the authors discuss how these data can be helpful to advance our understanding of the topic under study?

-Is public health relevance addressed?

Reviewer #1: Yes the conclusions were discussed and presented. The public health significance can be elaborated

Reviewer #2: Conclusions should be expanded and modified in line with the comments suggested, with a more detailed explanation of the relevance for public health management in the country and worldwide. One point requiring additional work, is being more explicit in what the impact of this work is for disease control, policy makers, and researchers.

**Editorial and Data Presentation Modifications?**

Reviewer #1: Minor revision with some more discussion on the ELISA idexx kit whether it is a OIE recommended kit with some relevant publications based on this kit. There can be a discussion why author preferred B cell medicated diagnostic rather than T cell medicated. The public health significance of the study need to be presented. Why the sero prevalence came less need to be elaborate whether hilly terrain or herd size or any other factor they consider to be important.

Reviewer #2: (No Response)

**Summary and General Comments**

Reviewer #1: Bovine tuberculosis-being am important animal and public health disease need to be studied. And as in Bhutan no previous studies of such is presented so it is important that such work should be carried out. Further tuberculin test based studies are also being encouraged by authors in there next project. The consumption of raw milk among public need to be also studied

Reviewer #2: The authors estimate the seroprevalence of bovine tuberculosis in Bhutan. This is the first study conducted in the country, so it provides a valuable information to get insights into the epidemiological situation of the disease in the area, surrounded by high prevalence countries. Overall, the conclusions of this article should be analyzed with caution. It is always complicated to estimate the prevalence of bovine tuberculosis in the absence of a gold standard, specially by using the ELISA test. It should be noted that this may be due to logistic reasons, which should be clearly mentioned along with the implications of it. I would like the authors to discuss a little more about the possible underestimation of the true prevalence induced here using ELISA test.

Thank you and congratulations for this very necessary and interesting work. Overall, this paper would benefit from some major revision, so I suggest some corrections and recommendations that may make it clearer and easier to read.

ABSTRACT

- I suggest adding a statement that the study was conducted in a region of Bhutan and did not include samples from across the country.

- Lines 27-28: “Although bTB is a zoonotic disease…”. This was already stated. Please avoid reiterating. 

- Line 31: “serum samples”?

- Is it possible to include a statement regarding the cattle census of which you randomly select samples?

- Lines 33-34: Please include information on the nature of the risk factors evaluated.

INTRODUCTION 

- Line 43: Although M. bovis represents the main causative agent of bTB, it may be caused by members of the Mycobacterium tuberculosis complex. Please rephrase.

- Lines 46-47: I suggest including a statement regarding other important aspects of the disease, such as latency and anergy, which add extra complexity to disease control.

- Line 50: Please change from “accounts” to “accounted”, as these are values from 2016. Additionally, could authors provide more up-to-date estimates?

- Line 52: Please add “may” before “get infected”. Are these contaminated aerosols derived from the close contact with infected animals? Please clarify.

- Lines 56-57: Please revise grammar and rephrase.

- Lines 57-59: In which countries? Could you provide further information that support this data? Any examples? In fact, there are several countries where, despite the efforts invested in bTB eradication through control programs (including developed countries), the disease is still endemic, and not negligible levels. Please correct.

- Lines 61-72: Please include references that support this statement (Lines 61-62). Additionally, there are some important risk factors missing (i.e., contact with other potential wild reservoirs, use of pastures leading to potential contact with other reservoirs -domestic and/or wildlife-, prevalence of the area,…). Please expand your explanation regarding this.

- Lines 69-70: I suggest adding other reference where the differences in the risk of bTB based on herd size are more evident (i.e., Conlan AJK, McKinley TJ, Karolemeas K, Pollock EB, Goodchild AV, et al. (2012) Estimating the Hidden Burden of Bovine Tuberculosis in Great Britain. PLoS Comput Biol 8(10): e1002730)

- Line 74: Please correct to “Single intradermal tuberculin (SIT) test”. Additionally, I suggest a adding some clarification of the limitations of the currently available diagnostic tests in terms of performance, as it is mentioned in Lines 179-181 (please see de la Rua et al., 2016 and Nunez-Garcia et al., 2018).

- Lines 77-78: I would not define it as “easy to apply” rather than ELISA provides a more standardized interpretation (i.e., it may be less subjective). Additionally, what about the sensitivity? Although it may be further discussed, it would be good to provide some context regarding why the ELISA is used in this epidemiological scenario.

- Line 80: but is it so because there is no bTB or rather it has not been tested yet? Any suspicions regarding bTB circulation in cattle population?

- Lines 81-85: Please include the year/range of years of these prevalence estimates, to provide a temporal context.

- 86-94: Just preventive measures or is it control and prevention measures? I suggest expanding the background and hypotheses of this study, maybe it would ease the reader to start with the statement presented in lines 88-91. Please clarify and rephrase.

MATERIAL AND METHODS

Overall, it would ease the reader to re-structure this section, maybe by including a lower number of subheadings (i.e., merge and re-structure sample size, sampling and sample collection subheadings) and specially by adding an expanded prior descriptive analysis of the cattle population. What is the population of which you take the sample? Proportion of herds out of the total cattle population in the region? Overall animals per herd? The sampling method is poorly presented. Was the sampling performed per district? How many herds per region/district? Additionally, the objectives should be clearly mentioned, along with the population under study, the outcome, and the statistical models applied.

- Lines 97-99: Do authors mean that Bhutan is subdivided into 4 regions that include 20 districts? One RLDC in the East, West, East Central and West Central? Please rephrase. Additionally, I suggest using another verb rather than “cater”.

- Line 99: Authors are assuming that the study was conducted in the eastern RLDC. Please add a clarification regarding the study area. I suggest something similar “The study was conducted in the East region, where the eastern RLDC provide veterinary services to six districts:…”

- Line 103: Please correct “owing to” to “due to”.

- Line 102: Please provide information on cattle census (i.e, number of cattle herds in the region, in each district), the population under study and from which you take the sample.

- Lines 105-110: Far too long a sentence. Please rephrase.

- Lines 110-111: Do authors mean that herds located in higher altitudes tend to practice stall-feeding, which are mainly crossbred dairy cattle? Please rephrase as it is confusing.

- Lines 111-113: Where? Across the region? In high altitudes? Please check grammar and rephrase. I suggest using the term shared pastures and watering points.

- Lines 116-117: This mention to the study area should be stated before, as suggested earlier. Additionally, I suggest “to determine the seroprevalence of bTB in cattle in Eastern Bhutan a cross-sectional study was conducted”.

- Line 118: With source of animal, did you mean herd of origin?

- Lines 122-129: please expand this section as stated before. What proportion of the total cattle herds/animals does 863 represent? Please expand the explanation on the coverage of the study samples (numerator) with respect to the cattle population of the region/district (denominator).

- Line 128: Please change to “Mycobacterium bovis”.

- Line 134: Please provide an extended descriptive analysis of the population, maybe extracted from this reference? Please include these descriptives at the beginning of the section and combine with information provided in Lines 132-141.

- Line 144: I suggest adding a mention earlier in Material and Methods regarding the year under study (2021, maybe in my comment of Lines 116-117?).

- Lines 146-147: Please move “was collected” at the end of the sentence.

- Lines 157-171: Material and Methods is a very long section. Have authors considered moving “Laboratory analysis” section to Supplementary material? Is it relevant to include all this information in the main manuscript?

- Line 175: In line with my previous comments, which descriptive data?

- Lines 178-179: Why were these values used? Could you provide any reference supporting these estimates?

- Lines 183-184: I would avoid using the term “known sensitivity and specificity of the test” as these are just estimates usually surrounded by a wide uncertainty. Please rephrase.

- Line 185: Please add reference and correct (Rogan and Gladen, 1978) same for line 190.

- Line 200: Age was categorized into three categories. Are these terciles? How was the distribution of data?

- Line 204: Please do not use abbreviations at the beginning of the sentence. 

- Lines 211-212: Results, please remove from Material and Methods.

- Lines 214-217: Maybe this section can be presented at the end of the manuscript.

RESULTS

- Overall, in this section, I suggest adding the “n” next to the % values (n = XX).

- Line 221-222: Please add “The maximum number of” before samples. Additionally, information on the cattle population per district is absent. 

- Lines 222-224: Actually, and in my opinion, Table 2 includes more descriptive data on the population under study than Table 1. Table 2 includes data on the estimated seroprevalence.

- Line 224: Please expand the title of the Table with the information included in it (i.e., number of samples, number of positive samples, etc.). Additionally, information in this table is scarce. It would be very useful to provide data on the number of samples per sub-district, along with the cattle census/herds in them (again, number of herds, number of tested herds, number of samples per herd, here in the Table or elsewhere). Otherwise, what is the point of including the Sub-districts column in the Table?

- Lines 227-228: Please provide data on the age distribution (i.e., median/mean age, interquartile range), and if a statistical test was used to evaluate these differences between young, adult and old and the outcome variable it should be stated, if not, please include it.

- Lines 228-229: Please check grammar. 

- Line 238: Please add %.

- Lines 238-242: It would be nice to include a graph

---

## [Decision Letter · Decision Letter 1]

17 May 2024

Dear Dr Wangmo,

We are pleased to inform you that your manuscript 'Seroprevalence and risk factors associated with bovine tuberculosis in cattle in Eastern Bhutan' has been provisionally accepted for publication in PLOS Neglected Tropical Diseases.

Best regards,

Husain Poonawala

Academic Editor

Stuart Blacksell

Section Editor

Reviewer's Responses to Questions

**Key Review Criteria Required for Acceptance?**

**Methods**

-Are the objectives of the study clearly articulated with a clear testable hypothesis stated?

-Is the study design appropriate to address the stated objectives?

-Is the population clearly described and appropriate for the hypothesis being tested?

-Is the sample size sufficient to ensure adequate power to address the hypothesis being tested?

-Were correct statistical analysis used to support conclusions?

-Are there concerns about ethical or regulatory requirements being met?

Reviewer #1: The methodology is updated and correct

**Results**

-Does the analysis presented match the analysis plan?

-Are the results clearly and completely presented?

-Are the figures (Tables, Images) of sufficient quality for clarity?

Reviewer #1: The results and analysis are clearly presented

**Conclusions**

-Are the conclusions supported by the data presented?

-Are the limitations of analysis clearly described?

-Do the authors discuss how these data can be helpful to advance our understanding of the topic under study?

-Is public health relevance addressed?

Reviewer #1: The conclusions are supported clearly

**Editorial and Data Presentation Modifications?**

Reviewer #1: Accept

**Summary and General Comments**

Reviewer #1: The revision was updated and modified as suggested

PLOS authors have the option to publish the peer review history of their article (what does this mean?). If published, this will include your full peer review and any attached files.

Reviewer #1: No

---

## [Editor Report · Acceptance letter]

22 May 2024

Dear Dr Wangmo,

We are delighted to inform you that your manuscript, "Seroprevalence and risk factors associated with bovine tuberculosis in cattle in Eastern Bhutan," has been formally accepted for publication in PLOS Neglected Tropical Diseases.

Best regards,

Shaden Kamhawi

co-Editor-in-Chief

Paul Brindley

co-Editor-in-Chief
